# *Eucalyptus pellita* Coppice vs. Seedlings as a Re-Establishment Method in South Sumatra, Indonesia

**Eko B. Hardiyanto** [1,*]**, Maydra A. Inail** [2]**, Daniel S. Mendham** [3]**, Erlanda Thaher** [2] **and Benardo K. Sitorus** [2]

1 Faculty of Forestry, Gadjah Mada University, Yogyakarta 55281, Indonesia
2 PT. Musi Hutan Persada, Muara Enim, Prabumulih 31172, Indonesia; maydra-al@jpn.mhp.co.id (M.A.I.); erlandathaher91@gmail.com (E.T.); benardokristian@gmail.com (B.K.S.)
3 CSIRO Land and Water, GPO Box 1700, Canberra, ACT 2601, Australia; daniel.mendham@csiro.au
* Correspondence: ebhardiyanto@ugm.ac.id

**Abstract:** *Eucalyptus pellita* can be regenerated through coppice. We report on the first known study of full-rotation productivity of *E. pellita* coppice and seedling re-establishment methods. We conducted this study at a high productivity site in South Sumatra, with the objectives to (1) evaluate the productivity of a first rotation of coppice stand in comparison with a replanted seedling crop, (2) investigate the effect of nitrogen (N) and phosphorus (P) fertiliser application on growth, and (3) examine the effect of timing of coppice stem number reduction on growth. The experiment was laid out in a randomised complete block design replicated five times. At the end of rotation (6 years after establishment) the coppice stand had substantially higher productivity (height 23.7 m, diameter 16.4 cm, volume 269.9 $m^3$ $ha^{-1}$, and MAI 45.0 $m^3$ $ha^{-1}$ $y^{-1}$) compared to the replanted seedling stand (height 20.5 m, diameter 13.8 cm, volume 193.6 $m^3$ $ha^{-1}$, and MAI 32.3 $m^3$ $ha^{-1}$ $y^{-1}$). Coppice stand also had higher biomass production but slightly lower wood density than the seedling stand. Neither stand responded to application of N fertiliser, and only the seedling stand responded positively to P fertiliser addition. Coppice reduction to one stem at 2 months after tree felling produced the stand with slightly greater diameter than that at 4 months after tree felling, but had approximately the same volume.

**Keywords:** *Eucalyptus pellita* growth; coppice regeneration; coppice reduction; fertiliser application

## 1. Introduction

*Eucalyptus pellita* plantations have been grown in Sumatra and Kalimantan, Indonesia since the early 2000s for pulpwood production; it has replaced *Acacia mangium* as the key plantation species, in response to severe losses of acacias due to *Ceratocystis* fungal wilt disease [1]. To date, all plantations have been grown using seedlings or clonal material; no coppice plantations have ever been attempted.

Like several other *Eucalyptus* species, *E. pellita* has an ability to resprout from the cut stump via dormant epicormic buds under the live bark. Therefore, *E. pellita* plantations can be re-established through coppice regrowth. Coppice eucalyptus plantations have been practiced for a number of eucalypt plantation species with a great deal of success in many parts of the world, such as South Africa [2–4], Brazil [5–8], Australia [9], and China [10]. The productivity of a coppice plantation has been reported to be similar or greater than that of the originally planted crop [4,5,8,11], but there may be a trade-off in productivity if survival (and therefore residual stocking) is low, vigour is less, or if the genetic material has improved markedly.

This study is the first, to our knowledge, that reports on the productivity of a coppice plantation of *E. pellita* compared to a seedling re-establishment method. The specific objectives of the study were to assess: (1) survival and growth of coppice plantation compared to seedling plantation, (2) the growth response of the coppice stand to the

application of nitrogen (N) and phosphorus (P) fertiliser, and (3) the growth response to the timing of coppice reduction.

## 2. Materials and Methods

### 2.1. Site Description and Treatment

The study was located in Gemawang, South Sumatra (03°64′25″ S, 104°00′53″ E) at an elevation of 100.2 m above sea level. This region experiences a tropical climate. During the experimental period, a nearby meteorological station recorded average daily temperatures of 29 °C, with mean annual maximum and minimum temperatures of 23 °C and 32 °C, respectively. Mean annual rainfall was 2522 mm, with 78% of this falling in the 7 months between October and April. Mean relative humidity was 77%.

The experimental site has a flat topography with Ultisol soils [12] derived from sedimentary rock consisting of sandy tuff, sandstone, and clay stone. Selected soil properties (0–10 cm) before the trial establishment were: bulk density 0.92–0.94 g cm$^{-3}$, pH (H$_2$O) 4.16, clay 55.0%, silt 10.2%, sand 34.8%, organic carbon 2.22%, total N 0.14%, Bray-P 0.79 mg kg$^{-1}$, exchangeable K 40.59 mg kg$^{-1}$, exchangeable Ca 69.86 mg kg$^{-1}$, and exchangeable Mg 14.68 mg kg$^{-1}$.

The first plantation was planted in October 2011 at a spacing of 3 m × 2 m (1666 trees ha$^{-1}$), using seedlings grown from seeds harvested from a seedling seed orchard of Muting provenance (Papua, Indonesia; 7°30′39″ S, 140°51′44″ E; altitude 38 m above sea level). The stand was selected for the trial as it had good stump stocking density (74.1% or 1235 trees ha$^{-1}$), and high productivity (124.0 m$^3$ ha$^{-1}$) when it was clear felled at age 4 years. The experiment was established in a factorial design, with two fertiliser treatments: N (0 and 250 kg N ha$^{-1}$) and P (0 and 30 kg P ha$^{-1}$). Both fertilisers were applied 2 months after clear-felling; N (in the form of Urea) was applied in a furrow between the rows of stumps, while P (in the form of triple super phosphate) was applied in pits around 15 cm from the stump. Two different times of coppice reduction (at 2 and 4 months after clear-felling) were carried out for all fertiliser treatment combinations, leaving the single most vigorous shoot per stump. Two shoots were left if there was a need to fill the gap as a result of an adjacent dead stump. The mean height of sprouts was 59.8 and 79.6 cm at 2 and 4 months after tree felling, respectively. The experiment was arranged in a randomised complete block design, with 16 measure trees per plot and one border row, replicated 5 times.

An experiment with replanted seedlings was also established at the same site, located next to the coppice trial at spacing of 3 m × 2 m (1666 trees ha$^{-1}$). Trees of the first rotation were harvested in September 2015, and seedlings of the second rotation were planted in November 2015. The seedlings were raised from seeds with similar genetic make-up to those used in the original planting of the first rotation, namely seeds from a seed orchard of Papua provenance. Similar fertiliser treatments and designs were applied as for the coppice trial, except the number of measure trees per plot was 25 and one border row.

Another experiment was carried out to further evaluate the growth of coppice compared to replanted *E. pellita* seedlings. The trial was established in May 2016 in South Sodong (a high productivity site). The original stand was planted at spacing of 3 m × 2 m (1666 trees ha$^{-1}$). There were two treatments: coppice regeneration and replanted with new seedlings. Both coppice and replanted stands were fertilized with P fertiliser at a rate of 30 P kg ha$^{-1}$ given at planting time. The experiment was laid out as a randomised complete block design, which was replicated 4 times. Each plot contained 30 measure trees with one row of border trees. Stump reduction was conducted at 6 months after the trees were cut using a similar method to the trial previously described in Gemawang.

### 2.2. Tree Growth and Biomass Assessment

The diameter at breast height (*D*) and total tree height (*H*) were measured annually until the end of rotation. The individual stem volume, *v*, was calculated as:

$$v = 0.25 \, \pi \, D^2 \, H \, F \qquad (1)$$

with a form factor (*F*) of 0.48, which had been derived for *E. pellita* [13]. The total volume per ha (*V*) for each treatment is the sum of the individual tree volume (*v*) per plot expressed on an areal basis.

Aboveground biomass production was estimated at the end of the rotation (age 6 years). Sixteen trees representing the range of diameter class were felled in the coppice stand. Biomass was separated into stem wood, bark, branches, and leaves. All tree components were weighed fresh in the field. Subsamples of each component were dried at 76 °C to a constant weight. The total dry weight of each biomass component was calculated by multiplying fresh weight with dry weight-fresh weight ratios [14]. Allometric equations were developed to describe the relationships between diameter at 1.30 m (breast height) and biomass components (Table 1), which were then applied to estimate aboveground biomass production at the plot level. The stem was divided into three sections of about equal length, and a 5 cm thick disc was taken from the middle of each section, then dried at 105 °C to a constant weight [15]. The wood density was calculated as a ratio between dry disc weight and disc green volume. The disc volume was determined by the water displacement method [16]. A similar procedure was used for the wood density estimation of the seedling crop.

**Table 1.** Allometric relationships between stem diameter and biomass components of coppice and seedling stands of *Eucalyptus pellita* at age six years.

| Component | Coppice | | Seedling * | |
|---|---|---|---|---|
| | Equation | $R^2$ | Equation | $R^2$ |
| Stem wood | $y = 0.0475 \, D^{2.7424}$ | 0.97 | $y = 0.005 \, D^{3.576}$ | 0.97 |
| Stem bark | $y = 0.0273 \, D^{2.2017}$ | 0.95 | $y = 0.005 \, D^{2.799}$ | 0.90 |
| Branches | $y = 0.0543 \, D^{1.6686}$ | 0.82 | $y = 0.135 \, D^{1.560}$ | 0.81 |
| Leaves | $y = 0.0032 \, D^{2.5274}$ | 0.98 | $y = 0.631 \, D^{0.918}$ | 0.72 |

* from Inail et al. (2019) [17].

### 2.3. Data Analysis

An analysis of variance (ANOVA) was performed to determine significant differences between treatments. R.ver. 4.11 software was used for all statistical analysis.

## 3. Results and Discussion

### 3.1. Coppice vs. Seedling Stands

After felling of the original stand, the tree survival was high (1270 trees ha$^{-1}$), which was more than the critical level of 1000 stumps ha$^{-1}$ suggested by Schonau [2] to be required for successful establishment of coppice plantations. The coppice stand of *E. pellita* grew significantly faster than the replanted seedling stand in all measured growth characteristics at the age of one, three, and six years, despite the fact that both regeneration methods used the same genetic planting material (Table 2). At age six years, the survival rate of the coppice stand was high (85.3% of the initial sprouting stump or 1170 stumps ha$^{-1}$), while the survival rate of the replanted seedling stand was lower (67.1% or 1119 trees ha$^{-1}$) due to waterlogging that occurred during the first six months after planting in some parts of the plots.

**Table 2.** Growth of the coppice and replanted seedling stand of *Eucalyptus pellita*.

| Regeneration Method | Mean Height (m) | Mean Diameter (cm) | Volume (m³ ha⁻¹) |
|---|---|---|---|
| *Age 1 year* | | | |
| Coppice | 7.9 ± 0.10 | 7.5 ± 0.05 | 22.6 ± 1.07 |
| Seedling | 5.0 ± 0.08 | 4.4 ± 0.08 | 5.6 ± 0.43 |
| *p-value* | *<0.0001* | *<0.0001* | *<0.0001* |
| *Age 3 years* | | | |
| Coppice | 14.9 ± 0.24 | 13.1 ± 0.25 | 123.9 ± 5.69 |
| Seedling | 12.8 ± 0.15 | 11.0 ± 0.16 | 75.6 ± 7.24 |
| *p-value* | *0.0009* | *0.0002* | *0.0003* |
| *Age 6 years* | | | |
| Coppice | 23.7 ± 0.40 | 16.4 ± 0.32 | 269.9 ± 26.16 |
| Seedling | 20.5 ± 0.36 | 13.8 ± 0.15 | 193.6 ± 19.81 |
| *p-value* | *0.003* | *0.001* | *0.0004* |

At three years of age, the coppice stand had almost double the stem volume of the seedling stand, which was statistically significant ($p = 0.0003$). The mean height, diameter and stem volume of the coppice stand were 14.9 m, 13.1 cm, and 123.9 m³ ha⁻¹, respectively, whereas the corresponding value of seedling stand was 12.8 m, 11.0 cm, and 75.6 m³ ha⁻¹, respectively. While the absolute stand productivity difference between the two stands became larger, the proportional difference in stand productivity between the coppice and replanted seedling stand declined with age. At the end of rotation (age six years), the mean height, diameter, and stem volume of the coppice stand were 23.7 m, 16.4 cm, and 269.9 m³ ha⁻¹, respectively, whereas the corresponding value of seedling stand was 20.5 m, 13.8 cm, and 193.6 m³ ha⁻¹, respectively. All three growth variables were significantly different between the two stands at age six years. The proportional stem volume difference between the two stands decreased from 303% at age one year to 64% at age three years, and to 39% at the rotation end of six years (Table 2). Whilst others have noted a declining growth rate with age in coppiced *Eucalyptus* [4,18], our observation was that coppice stands maintained a substantially higher current annual increment (CAI) than seedling stands, for the full rotation. By age three, the coppice had gained a 48 m³ ha⁻¹ advantage over the seedling crop, suggesting that coppice reaches peak production one to two years earlier than seedlings, and the additional wood that is gained before the seedlings catch up is maintained through to the end of the rotation (Figure 1).

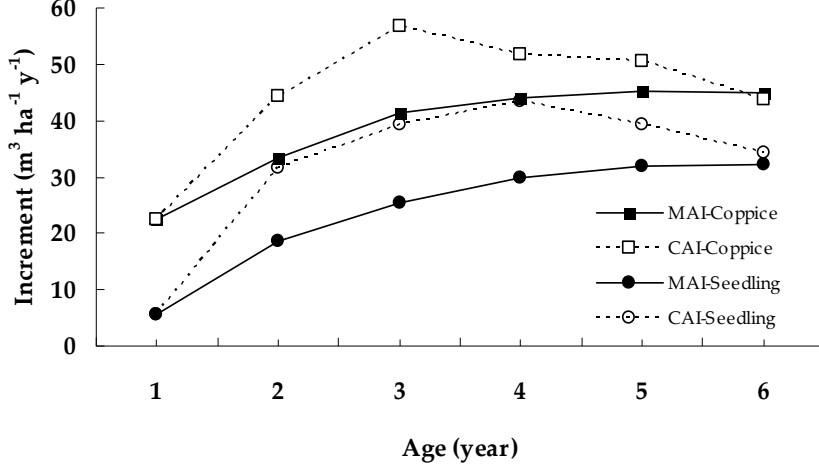

**Figure 1.** Mean annual increment (MAI) and current annual increment (CAI) of coppice and seedling stand of *Eucalyptus pellita*.

Higher productivity of the coppice stand compared with the seedling stand was also apparent from their wood biomass production. At six years after plantation establishment,

the total wood biomass production of the coppice stand, at 161 Mg ha$^{-1}$, was significantly higher ($p = 0.001$) than that of the seedling stand at 124 Mg ha$^{-1}$ (Figure 2). Stem wood contributed the highest proportion to the total biomass production in both stands, namely 81 and 79%, respectively, for coppice and seedling stands.

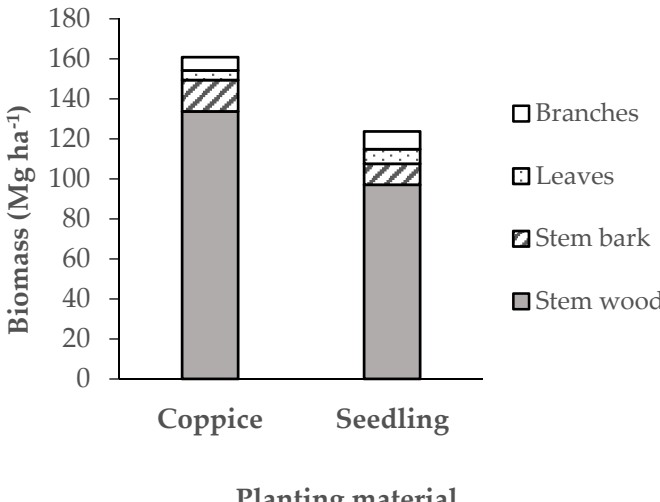

**Figure 2.** Biomass production of *Eucalyptus pellita* from different planting materials at age six years.

The wood density of the coppice stand ranged from 509 to 626 kg m$^{-1}$ (mean, 563 kg m$^{-1}$), which was slightly lower than that of the seedling stand of the same age of six years old, which ranged from 543 to 652 kg m$^{-1}$ (mean, 583 kg m$^{-1}$) (unpublished company data). Other studies with *Eucalyptus* have found a slightly lower wood density of coppice compared to seedlings. For example, coppiced *Eucalyptus grandis* × *E. urophylla* plantations in South Africa had wood density of 515 to 532 kg m$^{-1}$, while the seedling parent stand ranged from 545 to 563 kg m$^{-1}$ [19]. In *E. grandis* the wood density was reported to be 416 and 433 kg m$^{-1}$ for coppice and seedling stand, respectively [2]. The lower wood density of these coppice stands is likely due to its faster growth (and access to more water through the existing root system), compared to the seedling stand.

The second coppice vs. seedling experiment at South Sodong also had higher productivity of the coppice stand compared to the replanted seedling stand. The mean height, diameter, and volume of the coppice stand was 20.5 m, 14.5, and 246.3 m$^3$ ha$^{-1}$, respectively, while the mean height, diameter, and volume of the replanted seedling stands was 19.6 m, 14.1 cm, and 212.7 m$^3$ ha$^{-1}$, respectively, at age six years. The coppice stand had a 33.6 m$^3$ (15.8%) higher stem volume compared to the replanted seedling stand at the rotation end of six years. The survival rate of the coppice stand was 85.7% at age six years, while for the seedlings it was 67.1%.

Our findings that coppice productivity exceeded that of seedlings is supported by a number of studies with coppiced eucalypt plantations. A series of trials with coppice stands of *E. grandis* × *E. urophylla* in South Africa had slightly higher (albeit not significant) volume production at a high-productivity site and a much higher yield at low-productivity sites, compared with those of seedling crops [4]. At their lower productivity site, the difference between coppice and replanted stand was 288% at age one year, which reduced to 44% by rotation end at eight years. A number of reasons have been proposed as to why coppice stands have faster growth than replanted seedling stands. Young coppice shoots may be able to obtain nutrients and water from a larger soil volume, as their rooting system had already been developed [20,21]; also, it has a higher root-leaf ratio and lower belowground resource allocation, due to its existing, established root system [11]. Coppice stands are able to occupy the site rapidly, leading to higher total leaf area at an earlier phase of stand development. In addition, unlike seedling stands, coppice stands do not experience establishment shock [22].

Compared with the original seedling crop, the coppice stand in our study had a higher stem volume (46% higher) at the common age of four years; the stem volume was 124.0 and 181.5 m$^3$ ha$^{-1}$ for the original seedling and coppice stand, respectively. Similar results were found in a fertiliser study on a coppice stand of an *Eucalyptus urophylla* hybrid in Brazil, in which the coppice stand yielded 53.9% higher in stem volume than the original-first rotation crop at the common age of five years [8].

### 3.2. Responses to N and P Fertiliser Application

At age three years, the effect of N (250 kg N ha$^{-1}$) on growth of the coppice stand was not significant (height, $p = 0.83$ diameter, $p = 0.33$; volume, $p = 0.07$), likewise the addition of P fertiliser had no significant effect on growth (height, $p = 0.47$; diameter, $p = 0.95$, volume, $p = 0.67$). The interactions between N and P fertiliser application on growth were also not significant (height, $p = 0.27$; diameter, $p = 0.15$; volume, $p = 0.21$). While height and diameter were clearly not impacted by fertilizer application (Table 2), there was a trend for an additional 10–14 m$^3$/ha of standing volume (4–5% increase) in both of the fertilized treatments at age six years (Table 2), which is worthy of further exploration in future.

The lack of significant response to N and P fertiliser application is similar to findings in other studies with *Eucalyptus* coppice plantations. For example, no growth response was found to the application of N (100 kg N ha$^{-1}$) and P (40 kg P ha$^{-1}$) in *Eucalyptus urophylla* × *E. grandis* coppice stands at two sites in Brazil [7]. Similarly, no significant growth response to N fertiliser addition (94 kg N ha$^{-1}$) was found in a *Eucalyptus grandis* coppice in South Africa [3]. Another study on the application of N fertiliser in a coppiced *E. urophylla hybrid* crop found no positive growth response [8]; however, the coppice stand in that study did respond strongly to added K fertiliser, as K is the most limiting nutrient in Brazilian soils [7,8].

At age three years, the effect of N fertiliser addition (250 kg ha$^{-1}$) in a newly planted seedling stand was also not significant for any growth parameters (height, $p = 0.26$; diameter, $p = 0.22$; volume, $p = 0.82$). Similarly, the effect of N was not significant on growth at age six years. In contrast, the addition of P fertiliser had significantly increased growth at age three years (height, $p = 0.029$; diameter, $p = 0.002$; volume, $p = 0.006$), and the responses remained significant through to rotation end at age six years (height, $p = 0.019$; diameter, $p = 0.018$; volume, $p = 0.025$). Interactions between N and P fertiliser application were not significant for any growth parameters. Without added P, the mean height, diameter, and volume were 17.7 m, 13.3 cm, and 154.4 m$^3$ ha$^{-1}$, respectively, while, with added P (30 kg P ha$^{-1}$), the mean height, diameter, and volume were 18.8 m, 14.6 cm, and 185.0 m$^3$ ha$^{-1}$, respectively. The responsiveness of this plantation to P but not N supports the findings from other studies in the region that P is important at planting but not N [1,17]. The current practice of fertiliser regime for establishing seedling plantation of *E. pellita* in the region is to add P fertiliser at a rate of 24 kg P ha$^{-1}$, given at planting time. No N fertiliser is applied; however, it may be required in future rotations to obtain optimal growth [17].

### 3.3. Responses to Coppice Reduction

At age three years, the timing of coppice reduction had a significant effect on diameter ($p = 0.004$), but not on height ($p = 0.97$) or volume ($p = 0.35$). At age three years, the mean height, diameter, and stem volume for the coppice that had been reduced at two months after stem felling was 14.8 m, 13.5 cm, and 103.2 m$^3$ ha$^{-1}$, respectively, while for coppice that had been reduced at four months after tree-felling, they were 14.8 m, 12.7 cm, and 108.5 m$^3$ ha$^{-1}$, respectively. Similarly, at age six years, the timing of coppice reduction was still significant for diameter ($p = 0.024$), but not for height ($p = 0.51$) or stem volume ($p = 0.72$). At age six years, the mean height, diameter and volume for the coppice reduction treatment at two months after stem-felling was 23.9 m, 16.8 cm, and 255.5 m$^3$ ha$^{-1}$, respectively, while the for the reduction at four months after tree felling, they were 23.6 m, 15.9 cm, and 254.3 m$^3$ ha$^{-1}$, respectively. A study on a coppice trial of *Eucalyptus grandis* × *E. camadulensis* [23] found that early coppice reduction (at a shoot

height of 2 m) did not reduce growth significantly, compared with that of a late coppice reduction (at a shoot height of 4 m). Results of this present study suggest that early coppice reduction at two months (mean shoot height of 0.6 m) after stem-felling should be practiced for establishing coppice plantations of *E. pellita* as it is operationally much easier to do and has cost savings, as shoots are shorter as well as conveying a small benefit to stand growth. The risk of windthrow for coppice reduction conducted at two months was very low, much lower than coppice reduction carried out at four months.

### 3.4. Future Prospects of Coppice Plantation

Providing that there is a high initial stump stocking (>1000 stumps ha$^{-1}$), a coppice plantation of *E. pellita* provides an alternative regeneration method for plantation establishment, as it is able to produce a higher yield more inexpensively, at least in the first rotation after a seedling crop. Coppice plantation is worth considering for establishment in remote areas, where access to the sites is difficult and costly. Coppicing may be a better option at low productivity sites [1], where the low cost of establishment and higher productivity will help the grower to obtain better financial returns. Further work is warranted on low productivity sites in this region, which are typically characterised by a shallow depth to the impeding (plinthite or glayed) layer (<50 cm) and a high clay content (>50%) [1,24]. It is worth noting that coppice crops also require less soil disturbance, so are likely to help maintain the productivity of the site in the longer term if minimal site preparation can be practiced [25].

Coppice plantations are unlikely to be prospective whenever the stump stocking is low (<1000 trees ha$^{-1}$). Availability of planting materials with higher wood production (i.e., higher yielding clones) than the current material will also be a factor that needs to be considered prior to establishing the next rotation; in some cases, newer genetic material may still outperform coppice, but again this needs further work before a definitive answer can be provided.

## 4. Conclusions

The first rotation of coppice stands of *E. pellita* had faster growth and greater stem volume than replanted seedling crops of the same genetic planting material, with an additional 60 m$^3$/ha harvested from the coppice plots at our high productivity site at age six years, compared to the adjacent seedling plots. N and P fertiliser applied to the coppice stand had no significant effect on growth, contrasting with the adjacent seedling crop, which did significantly respond to P fertilizer at establishment. Early coppice reduction (at two months) tended to give slightly better diameter productivity compared to reduction at four months and is easier to implement operationally; therefore, it is the recommended practice going forward.

**Author Contributions:** Conceptualization, D.S.M., E.B.H., and M.A.I.; methodology, E.B.H. and D.S.M.; validation, E.B.H., M.A.I., and D.S.M.; formal analysis E.B.H. and M.A.I.; investigation, E.B.H., M.A.I., D.S.M., E.T., and B.K.S.; data curation, M.A.I., E.B.H., E.T., and B.K.S.; writing—original draft preparation, E.B.H. and M.A.I.; writing—review and editing, D.S.M.; project administration, M.A.I., E.B.H., and D.S.M.; funding acquisition, D.S.M., E.B.H., and M.A.I. All authors have read and agreed to the published version of the manuscript.

**Funding:** This research was financially supported by the Australian Centre for International Agricultural Research (ACIAR) through ACIAR Project FST/2014/064, and PT. Musi Hutan Persada, South Sumatra.

**Acknowledgments:** The authors wish to acknowledge the Australian Centre for International Agricultural Research (ACIAR) for funding the study through ACIAR Project FST-2014-064, and the Management of PT. Musi Hutan Persada for their commitment to its implementation and support. We would also like to thank the manager of the R&D of PT. Musi Hutan Persada (Bambang Supriadi) and Silviculture Team at the R&D of PT. Musi Hutan Persada for their field assistance with the establishment, maintenance, and measuring of the trials.

**Conflicts of Interest:** The authors declare no conflict of interest.

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
