# Peer review of "Eucalyptus pellita Coppice vs. Seedlings as a Re-Establishment Method in South Sumatra, Indonesia"

_forests, doi:10.3390/f13071017_

Round 1

Reviewer 1 Report

I believe that the manuscript has merit to be published.

Some considerations:

Abstract:

-how many sprouts per tree? 

- insert MAI

MM:

Line 61 - The first plantation was planted in… when? Month and year 

Results and discussion

Is it possible to insert a graph with the mean and current annual increment?

A little strange is the high difference in productivity at the end of the study. In the initial phase it is common, but with older ages (tend to have little difference), it draws a lot of attention. I believe this is a point to reinforce the discussion.

Another point is the Wind risk with coppice reduction at 2 months. insert something about it

And the fertilization inserts some discussion about the eucalypt nutrient budget in one rotation. 

Author Response

I believe that the manuscript has merit to be published.
Some considerations:
Abstract:
-how many sprouts per tree? Leaving one stem (sprout) per tree after the sprout reduction as already described in the abstact
- insert MAI It has been added
MM:
Line 61 - The first plantation was planted in… when? Month and year Planting month and year have been
included
Results and discussion
Is it possible to insert a graph with the mean and current annual increment? A graph showing MAI and
CAI has been added as suggested (Figure 1).
A little strange is the high difference in productivity at the end of the study. In the initial phase it is
common, but with older ages (tend to have little difference), it draws a lot of attention. I believe this is a
point to reinforce the discussion. It is true that the absolute difference in productivity (volume) between
coppice and seedling stand was high at the end of rotation, it became larger towards the end of rotation.
However, the proportional difference in volume between two stands declined with age. For an example,
at age 1 year the proportional volume difference between the two stands was 303% , decreased to 64%
at age 3 years and to 39% at the end of rotation (age 6 years).
Another point is the Wind risk with coppice reduction at 2 months. insert something about it The wind risk
of coppice stand with coppice reduction at 2 months was quite low, much lower than that at 4 months.
This has now been noted in the revised manuscript.
And the fertilization inserts some discussion about the eucalypt nutrient budget in one rotation. It has
been added (line 239-242).

Reviewer 2 Report

Add and quote used methodology or norms to calculate dry biomass.

In table 1 missing statistics to see if there is or there is no difference between the three research variables by age. That should be added.

Figure 1 should be repaired with legends. There is too little cited literature for this type of research and this journal.

I do not support scientific papers without statistical analyzes in research results.

Author Response

Add and quote used methodology or norms to calculate dry biomass. Has been added as
suggested
In table 1 missing statistics to see if there is or there is no difference between the three research
variables by age. That should be added. The statistical tests have beed added
Figure 1 should be repaired with legends. There is too little cited literature for this type of research
and this journal. Thank you for spotting the missing legend; it has been revised accordingly
I do not support scientific papers without statistical analyzes in research results. Thanks for
suggestion. All statistical analyses in research results have been provided. .

Round 2

Reviewer 2 Report

All my remarks have been corrected.